# Exploring *Micromonospora* as Phocoenamicins Producers

**DOI:** 10.3390/md20120769

**Published:** 2022-12-07

**Authors:** Maria Kokkini, Cristina González Heredia, Daniel Oves-Costales, Mercedes de la Cruz, Pilar Sánchez, Jesús Martín, Francisca Vicente, Olga Genilloud, Fernando Reyes

**Affiliations:** Fundación MEDINA, Centro de Excelencia en Investigación de Medicamentos Innovadores en Andalucía, Parque Tecnológico Ciencias de la Salud, Avda. del Conocimiento 34, 18016 Armilla, Granada, Spain

**Keywords:** antimicrobial resistance, drug discovery, natural products, actinomycetes, *Micromonospora*, polyketides, spirotetronates, phocoenamicins

## Abstract

Over the past few years, new technological and scientific advances have reinforced the field of natural product discovery. The spirotetronate class of natural products has recently grown with the discovery of phocoenamicins, natural actinomycete derived compounds that possess different antibiotic activities. Exploring the MEDINA’s strain collection, 27 actinomycete strains, including three marine-derived and 24 terrestrial strains, were identified as possible phocoenamicins producers and their taxonomic identification by 16S rDNA sequencing showed that they all belong to the *Micromonospora* genus. Using an OSMAC approach, all the strains were cultivated in 10 different media each, resulting in 270 fermentations, whose extracts were analyzed by LC-HRMS and subjected to High-throughput screening (HTS) against methicillin-resistant *Staphylococcus aureus* (MRSA), *Mycobacterium tuberculosis* H37Ra and *Mycobacterium bovis*. The combination of LC-UV-HRMS analyses, metabolomics analysis and molecular networking (GNPS) revealed that they produce several related spirotetronates not disclosed before. Variations in the culture media were identified as the most determining factor for phocoenamicin production and the best producer strains and media were established. Herein, we reported the chemically diverse production and metabolic profiling of *Micromonospora* sp. strains, including the known phocoenamicins and maklamicin, reported for the first time as being related to this family of compounds, as well as the bioactivity of their crude extracts. Although our findings do not confirm previous statements about phocoenamicins production only in unique marine environments, they have identified marine-derived *Micromonospora* species as the best producers of phocoenamicins in terms of both the abundance in their extracts of some major members of the structural class and the variety of molecular structures produced.

## 1. Introduction

Antimicrobial resistance (AMR), the ability of microorganisms to progressively become resistant to known antimicrobials, is an urgent health challenge worldwide and can be considered a hidden global pandemic [1]. Historically, natural products (NPs) have played a major role in the discovery of antimicrobials. They have provided the pharmacophores for most of the major classes of antibiotics, including the *β*-lactams, aminoglycosides, macrolides, tetracyclines, rifamycins, glycopeptides, streptogramins, and lipopeptides [2]. In fact, according to Newman and Cragg’s review analyzing the new drugs approved from January 1981 to September 2019, out of all the new antibiotics (vaccines excluded), 68.5% were natural products or natural product derivatives, while only 27.7% were totally synthetic antibiotics [3].

Natural products (NP) are structurally perfected by evolution to deliver particular biological functions [4] and are therefore an invaluable source of inspiration in antibiotic design and development [5], compared to the lack of success of synthetic molecules, which often miss the physiochemical properties needed to penetrate bacterial membranes [6]. These special characteristics in comparison with the synthetic compounds carry both advantages and challenges in the discovery of new antibiotic leads [4].

The challenges include various shortfalls and technical barriers, such as the inability to culture the majority of environmental microorganisms in the laboratory, the time-consuming dereplication process, and obtaining the optimal conditions to scale up their production and acquire a sufficient amount to isolate the active principles and elucidate their structures [4,7]. Since the 1990s and onwards, these challenges, along with other economic issues have led to the withdrawal of large pharmaceutical companies from antibiotic research and directed them towards other indications. As a result, there is an absence of new classes of antibiotics reaching the market, leaving antibiotic discovery and development to academic and governmental institutions, and small biotech companies [6,8,9].

Over the past few years, the development of technological and scientific advances such as improved analytical tools, genome mining and engineering, and microbial culturing advances, such as co-cultivation and in situ cultivation [10], are opening up new opportunities to explore new microbial sources and make NP drug discovery attractive again [4,7]. Metabolomics was developed as an approach for metabolic profiling and can provide information on the metabolite composition in NP extracts, thus helping to prioritize NP extracts for isolation and accelerate dereplication. Along with the one strain, many compound (OSMAC) approach, metabolomics can be used to explore and statistically validate relationships between culture parameters and chemical diversity in a microbial isolate. In this respect, the Global Natural Products Social (GNPS) molecular networking platform [11] can organize thousands of sets of MS/MS data recorded from a given set of extracts and visualize the relationship of the metabolites as clusters of structurally related molecules, annotating unknown analogues and new NP scaffolds [2,4].

Spirotetronates are a class of polyketides produced by actinomycetes, most of them belonging to the genera of *Streptomyces*, *Micromonospora* and *Actinomadura* [12,13]. The producing microorganisms have been isolated from many different sources, such as soil, plants, sea and marine sediments and organisms, and are geographically distributed worldwide. They have demonstrated a wide variety of biological activities, mostly antibiotic, against gram-positive bacteria and antitumor [12,13,14]. Despite their potent bioactivities, their structural complexity and the difficulties towards large-scale culturing, have been limiting their availability for further preclinical and clinical studies [3].

Recently, the spirotetronate class has grown with the discovery of the phocoenamicin family of compounds, some of them isolated in our laboratory [15,16]. In this work, a total of 27 actinomycete strains identified as possible phocoenamicins producers using our LC/MS analytical platform [17], were cultured using an OSMAC approach in order to select the most promising ones, as well as fermentation media for the production of members of this structural class. The analysis of the crude microbial extracts revealed that they produce several related compounds not disclosed before, and that their production is dependent on the culture medium. Herein, we reported the diverse metabolic production of these strains, as well as the bioactivity of their crude extracts against methicillin-resistant *Staphylococcus aureus* (MRSA), *Mycobacterium tuberculosis* H37Ra and *Mycobacterium bovis.*

## 2. Results

### 2.1. Taxonomy, Geography and Ecology of the Producing Microorganisms

The microbial strains used in this study belong to the culture collection of MEDINA foundation and were identified by searching an internal repository of LC/MS analyses of microbial extracts against an internal database of LC/UV/MS data of microbial natural products, which includes several members of the phocoenamicin family [17]. This search identified 27 microbial strains that potentially biosynthesize at least one of the phocoenamicins.

Taxonomic identification of the 27 strains was performed via sequencing of the 16S ribosomal gene with the universal primers fD1 and rP2. The sequences obtained were compared with the sequences of all prokaryotic species recognized as “type strains” with validly published names registered in the EzBioCloud database. The closest species, the (%) percentage of similarity, the source of isolation and the geographic origin of each strain are shown in Table 1. In the case of the strain CA-259211, it was not possible to obtain amplification product of the 16S ribosomal gene in any of the PCR conditions tested.

All the strains identified as phocoenamicin producers belong to the *Micromonospora* genus, suggesting that the common pathway for the production of these molecules must be highly conserved within species of this genus. The closest type species were distributed into seven different species, with *M. terminaliae* and *M. endophytica* getting assigned more than half of all the strains (18 strains), followed by *M*. *siamensis, M. chalcea, M. solis, M. chaiyaphumensis* and *M. aurantiaca.* Given that all three known phocoenamicins were originally isolated as a product of *Micromonospora* sp., and that all strains identified in this study as possible producers also belong to this genus, it can be proposed that the biosynthesis of this family of compounds might be confined to the *Micromonospora* species.

Despite the close taxonomic relation of the strains, they displayed great ecological and geographical diversity. Even though the phocoenamicins were originally isolated from marine-derived actinomycetes, the producers identified in this study were isolated from many different sources, as shown in Table 1. These sources include forest organic soil, either dry or humid, rhizosphere soil from various plants, rice cultivations, soil in swamp ecosystems, marine sediments and finally a strain isolated from a marine invertebrate. This demonstrates that the production of phocoenamicins is not attributed to the unique environmental conditions, as previously believed [18,19]. Regarding their geographical origin, they were widely distributed worldwide, covering almost all continents. In particular, the 27 strains were isolated from Costa Rica, Mexico, Spain, Georgia, Central African Republic, Comoros Islands and New Zealand, confirming the hypothesis that this family of compounds is highly conserved within the *Micromonospora* genus.

### 2.2. OSMAC Approach, Extraction and Metabolomics Analysis of the Fermentation Broths

The importance of growth conditions in the amount and diversity of secondary metabolite production is well recorded. Therefore, each of the 27 strains was cultured in 10 different media (One Strain Many Compounds-OSMAC approach) [20] to identify the best conditions and maximize their biosynthetic potential.

After 14 days of fermentation, the culture broths were extracted with acetone for subsequent liquid chromatography coupled to high resolution mass spectrometry (LC-HRMS) analysis using positive ionization mode. A total of 280 samples were obtained, 270 extracts from the fermentation broths and 10 extracts from the corresponding control media. Initially, a study of the production of the three known phocoenamicins was carried out in each of the conditions analyzed.

All 27 strains produced the three compounds in at least one of the fermentation media. For phocoenamicin, the (+)-HRESIMS data showed a peak at a retention time (RT) of 6.45 min displaying an ammonium adduct ion at *m*/*z* 1088.4982 [M + NH_4_]^+^, corresponding to the molecular formula C_56_H_75_ClO_18_. For phocoenamicin B, a peak with an adduct ion at *m*/*z* 1104.4927 [M + NH_4_]^+^, in agreement with the molecular formula C_56_H_75_ClO_19_ was detected at a RT of 5.62 min and for phocoenamicin C, a peak at a RT of 6.01 min with an adduct ion at *m*/*z* 1104.4933 [M + NH_4_]^+^, corresponding to the molecular formula C_56_H_75_ClO_19_. In all three cases, the characteristic UV spectrum of the phocoenamicins was observed and all the data were in good agreement with those previously published for the three compounds [15].

The manual exploration of the data suggested that phocoenamicin was the most abundant compound, detected in all strains and in almost every culture medium, with the exception of DEF-15, DEF-15S and DNPM for some of the strains. Phocoenamicin B was detected in lower amounts, but still significantly in all strains, and phocoenamicin C was the least produced metabolite and almost absent in extracts of some of the strains (CA-253038, CA-249379, CA-244669 and CA-108000).

Furthermore, the qualitative study of the production of the three phocoenamicins suggested that, in general for all 27 strains, the most suitable fermentation media to obtain these compounds were RAM2-P V2, FR23, M016, and SAM-6. On the other hand, APM9, FPY-2 and FPY-12 exhibited significantly lower production levels, while in DEF-15, DEF-15S and DNPM the production was very low or non-existent.

The extracts were further subjected to multivariate data analysis (MVDA) in order to explore the production potential of the strains in depth. The raw data obtained from the analysis were preprocessed by MZmine 2 [21] (version 2.53). The treatment included mass detection, chromatogram building and deconvolution, noise reduction and data filtering, peak detection and integration, chromatographic alignment and gap filling [21], resulting in a data matrix of 6050 metabolite features. The samples were organized in groups according to their taxonomy, geography, ecology, culture medium, and the different strains. The processed data were then converted into a suitable format (comma-separated values, CSV file) for each group and was uploaded to the MetaboAnalyst 5.0 platform [22] to carry out the statistical analysis. Missing values were replaced, and further feature filtering based on interquantile range (IQR) reduced the data to 2500 features. The purpose of the data filtering was to identify and remove variables that were unlikely to be of use when modeling the data. Normalization was performed using Pareto scaling and log transformation to make individual features more comparable and transform the data matrix into a more Gaussian-type distribution.

Principal Component Analysis (PCA) was used as an exploratory tool to reveal possible patterns, trends or outliers. PCA reduces data by projecting them onto lower dimensions that are called principal components (PCs). As it is an unsupervised chemometric analysis, it ignores any information about the different groups of the samples and can only distinguish classes when the within-group variation is sufficiently lower than the between-group variation. The aim is to create the best summary of the data while using the least PCs possible [23,24].

The samples were the extracts of the *Micromonospora* fermentations, and the features were peaks defined by an *m*/*z* value and RT. The extracts that only contained the different media without bacterial cultivation were used as control samples. Five different parameters were tested as class labels, in particular the different strains, taxonomic species, culture media, source, and geography and PCA was generated for each one of them to explore if these parameters can influence the metabolite production.

As shown in Figure 2, there was a close clustering of the control samples in all cases, demonstrating that the discrimination observed was mainly due to biological factors. PC1 and PC2 were capable of discriminating samples depending on the culture medium Figure 2e, with a total variance explained of 42.4%. Furthermore, the extracts in media that were previously identified as poor producing, were clustered closer to the control media. There was no detectable grouping based on neither of the other parameters Figure 2a–d. This suggests that such parameters do not contribute to a different chemical profile, since they do not influence the distribution of objects.

PCA was combined with partial least squares-discriminant analysis (PLS-DA) to confirm the previous observation and review in detail the chemical variation within the extracts. PLS-DA can be considered as a “supervised” view of PCA as it takes into account the different groups within the samples and maximizes the covariance between the data matrix and the class label. It can also be used for the selection of the most important features that drive the separation between the groups [25].

Since it is prone to overfitting and may create clusters even in random data, cross-validation (CV) and permutation tests are necessary steps when using PLS-DA. A significantly important model shows R_2_ (explained variance) and Q_2_ (predicted variance) values > 0.7 on cross-validation and *p*-value < 0.05 on the permutation test [26].

Only the Y-model that clusters the samples according to the culture medium could be used to explain chemical variation with values of R_2_ = 0.859 and Q_2_ = 0.841 and *p*-value < 0.01 on the permutation test (Figure 3a). Indeed, all the other PLS-DA attempts of the different Y-class models (strain, taxonomy, geography, and ecology) apart from the absence of discrimination (see Appendix A), resulted in overfitting and lack of validation. As a conclusion, the only parameter that could provoke a significant change in the metabolic profile of the different *Micromonospora* strains was the culture medium. As far as the control media were concerned, they were closely clustered for the media that were identified as potent phocoenamicin producers but scattered and close to all the extracts of the same medium for the poor producing ones, reaffirming the significant influence of the culture medium in the metabolite production of these strains.

Furthermore, the 3D loadings plot, as shown in Figure 3b, revealed the main compounds that were responsible for this separation. The components that were further from the center of the plot contributed the most to the separation. From the metabolites observed, only the ones that accounted for Variable Importance in Projection (VIP) higher than 1.5 were considered. Among them, the spirotetronate maklamicin was putatively identified via manual dereplication, as well as daidzein, genistein and the siderophores nocardamine and terragine B (see Section 2.4), confirming the chemical differences between the extracts cultivated in each medium.

As one of the most potent culture media, the RAM2-P V2 medium was selected for further analysis by liquid chromatography coupled to tandem mass spectrometry (LC-MS/MS) to improve its resolving ability and explore more profoundly the metabolic potential of the strains. Following the same procedure, the data obtained were processed with MZmine 2 and then analyzed using the MetaboAnalyst platform. The data matrix contained 1287 features and the samples were distributed into two groups according to their ecology: marine-derived, and terrestrial strains.

Hierarchical clustering was performed by MetaboAnalyst 5.0 to build a hierarchy of clusters according to the metabolic profile of the 27 strains. Hierarchical cluster analysis (HCA) is an unsupervised quantitative method used to assess the chemical similarity of different samples [27]. Ward’s clustering algorithm and Euclidean distance measure were used to create the dendrogram that is shown in Figure 4a. The three marine-derived strains were closely clustered indicating that they have a similar chemical response to the RAM2-P V2 medium. Moreover, the relative peak intensities of the three phocoenamicins were explored comparing the marine and terrestrial strains. As shown in Figure 4b, phocoenamicin was produced more within the terrestrial strains, while phocoenamicins B and C within the marine.

Lastly, keeping in mind the limitation of using non-quantitative mass spectrometry data, the analysis of the results allowed us to identify tendencies based on the peak area. The peak intensity plot of the processed data generated by MZmine 2 (Figure 5) suggested that the maximum levels of phocoenamicin production in RAM2-P V2 medium were obtained in extracts of the terrestrial strains CA-184181 and CA-238649, and for phocoenamicins B and C in the marine CA-214658 strain, followed by the also marine CA-218877, reaffirming the previous results that phocoenamicin is produced more by the terrestrial strains while phocoenamicins B and C by the marine-derived ones.

Moreover, as shown in Figure 5, the putative phocoenamicin derivatives identified (see Section 2.6) were also analyzed and their production levels were compared within the 27 strains in RAM2-P V2 medium. Among them, the compound with an adduct ion at *m*/*z* 920.4935 was produced more by the strains CA-248649, CA-244161 and CA-251294 and the compound with an adduct ion at *m*/*z* 774.4410 was produced more by the strain CA-249379. Lastly, the marine strains CA-214658 and CA-218877 demonstrated the highest diversity in producing these minor compounds, suggesting that they could be good candidates for the scale-up of the production and isolation of these putative minor analogues.

### 2.3. Antimicrobial Activity of the Crude Extracts

Antimicrobial activity of the crude extracts indicates the ability of the strains to produce metabolites of potential therapeutic interest [28]. The crude extracts of the 270 fermentations, as well as the 10 extracts from the control media, were subjected to high throughput screening against MRSA MB5393, *M. bovis* BCG and *M. tuberculosis* H37Ra to evaluate the antimicrobial potential of the strains and to study if a correlation could be established between antimicrobial activity and phocoenamicins production. The 10 extracts from the control media did not show any activity against the microorganisms tested, confirming that the bioactivity observed in the crude extracts can be exclusively attributed to the compounds produced by the *Micromonospora* strains and not to the fermentation media components. Figure 6 shows how bioactivity was influenced by the different fermentation media. The graph represents the average of the percentages of inhibition (%) obtained from each of the extracts from the 27 strains versus the different media in which they had been grown.

All 27 strains exhibited some type of antimicrobial activity under some of the conditions and several trends were observed. It was observed that in the activity (% inhibition) of the extracts, the medium in which the *Micromonospora* strains were grown was clearly a more determining factor than the different strains themselves. So, the highest inhibition against the three pathogens was observed in the extracts from the fermentation medium M016, followed by the fermentation media RAM2-P V2, SAM-6 and APM9. Comparing the three pathogens, the highest activity was detected against *M. tuberculosis* and then *M. bovis* and MRSA. To the contrary, the lowest inhibition was found in the extracts from fermentation media DEF-15, DEF-15S and DNPM. A correlation between these low levels of inhibition and the low levels of phocoenamicins produced in these media can be established based on the fact that the phocoenamicins have been described as compounds with antimicrobial properties against gram-positive pathogens [15]. However, the same correlation was not observed when comparing other fermentation media, where significant differences in phocoenamicins production, as in the case of FPY-12 and FR23 media, did not translate into significant differences in inhibition. Nevertheless, this could be attributed to synergism between the phocoenamicins and/or other bioactive compounds present in the *Micromonospora* extracts, since they contain complex mixtures. Lastly, these results revealed a rich antimicrobial potential within the 27 strains that should be further investigated. More assays should be performed against more pathogens in order to discover new antimicrobial activities and determine the responsible bioactive compound(s) in each case.

### 2.4. Dereplication of The Main Compounds Produced

The combination of the LC-UV-MS analyses, metabolomic profiling and molecular networking allowed the putative dereplication of the main compounds produced within the 27 strains in the different growing conditions, as shown in Table 2. The proposed dereplication was accomplished comparing the data against the in-house MEDINA database, the Dictionary of Natural Products (DNP), the COlleCtion of Open Natural ProdUcTs (COCONUT), the Natural Product Activity & Species Source database (NPASS) and the ChemSpider database. The comparative analyses led to the tentative identification of multiple, chemically diverse, compounds. The *m*/*z* detected, the corresponding adduct ion, the theoretical mass of each adduct and its deviation in ppm, the molecular formulae, and the RT are shown for each compound. Due to the detection of the molecules included in the table in multiple samples, representative values of *m*/*z* and retention time are displayed for each compound.

As mentioned above, the three phocoenamicins were putatively detected within all the extracts. Along with the spirotetronates of interest, maklamicin was also tentatively identified to be produced by all 27 strains. Maklamicin is also a spirotetronate first found to be produced by an endophytic *Micromonospora* sp. collected in Thailand and has shown strong to modest antimicrobial activities against various gram-positive bacteria [29]. As shown in Figure 7, structurally, maklamicin is closely related to phocoenamicins, having 11 carbons on its macrocycle and a decalin unit, as phocoenamicins. The closest proximity is with phocoenamicin B, their major differences being the absence of the two 6-deoxyglucose moieties, the 3-chloro-6-hydroxy-2-methylbenzoate moiety, and the diol side chain, unique in the phocoenamicins [15,29]. Within the 270 extracts of this study, the production of one of the two families conditioned the production of the other in all cases. Furthermore, phocoenamicin and maklamicin co-eluted in the crude extracts of the fermentation broths and further purification would be needed for their separation. 

Apart from the spirotetronates, the majority of the compounds putatively dereplicated belonged to the hydroxamate siderophores, and in particular to the desferrioxamine family, with the most common being deferoxamine [30], nocardamine [31], terragine B [32] and IC 202B [33]. The hydroxamate siderophores have therapeutic and diagnostic importance and have demonstrated antimicrobial and antitumor activities. Furthermore, antibiotics linked to siderophores, natural conjugates named sideromycins, have shown a higher antibacterial efficacy compared to normal antibiotics due to an enhanced uptake using the siderophore-mediated iron active transport and their use has been explored in a “Trojan horse” approach to overcome bacterial resistance to antibiotics [34,35]. As metal chelators, Desferal, the brand name of desferrioxamine B, was the first antibiotic used to remove excess iron in patients suffering from iron toxicity or overload, such as in the case of *β*-thalassemia patients that depend on blood transfusions [36].

Finally, various other compounds were tentatively identified within the extracts. The most abundant were the isoflavones daidzein [37], genistein [38] and glycitein [39] that have antioxidant, anticancer and antimicrobial properties [37,40] and several more metabolites with antibiotic properties, such as 21-demethyl-leptomycin A [41], that could explain the differences observed between the antimicrobial activity and the production levels of phocoenamicins in some of the extracts. The wide and diverse metabolic profile of these strains highlights their therapeutic potential and the need for further investigation.

### 2.5. Molecular Networking Analysis

Over the last few years, molecular networking (MN) has gained a lot of attention in the field of natural products. MN is a bioinformatic tool that explores chemical diversity visualizing the entire metabolome detected in a dataset. It is based on the assumption that chemistry influences how the molecules will be fragmented by tandem mass spectrometry. Therefore, structurally related compounds will have a similar fragmentation pattern and will be connected whereas the unrelated will be separated [42].

In a molecular network, each node represents a compound and the nodes with similar spectra are linked to form clusters or “molecular families”. Moreover, these links provide valuable structural information. Compounds that belong to the same cluster usually share the same core structure but differ due to simple transformations such as alkylation or oxidation. For example, a difference of 14 Da could suggest putative CH_2_ homologues and a difference of 16 Da an oxygenated analogue [43].

Molecular networking has led to the creation of Global Natural Products Social Molecular Networking (GNPS), a web-based platform (http://gnps.ucsd.edu, accessed on 2 December 2022) developed by researchers at the University of California at San Diego. GNPS uses an algorithm to compare the MS/MS spectra and is based on various parameters like the cosine score. After the analysis, searching of the GNPS libraries, in-house databases, and other public spectral libraries can lead to the rapid annotation of known compounds, making GNPS an important dereplication tool [11].

GNPS offers two main workflows, the classical MN and feature-based molecular networking (FBMN). In classical MN, raw MS/MS spectra files are processed directly to generate a MN. This often leads to multiple nodes for the same compound when detected over a large retention time span and creates huge MNs that are not fully representative of the actual number of compounds in the dataset. In order to overcome this problem, FBMN uses a feature detection and alignment tool, such as MZmine 2, to preprocess the raw data and is used for advanced molecular networking analysis, enabling the relative quantification of the compounds and resolution of isomers [44].

MN was used to organize the MS/MS data obtained from the crude extracts of the 27 strains grown in the medium RAM2-P V2 that was used as well as a control. Classical MN was dismissed because of the massive and difficult task to interpret the molecular network it created, with 2514 nodes organized in 65 clusters (at least two nodes connected). The results of the analysis are publicly available at: https://gnps.ucsd.edu/ProteoSAFe/status.jsp?task=37e734c22f5f43cab7f8983d5121cea6, accessed on 2 December 2022. Even though an overview of the metabolites and their relationships could be observed, FBMN workflow was chosen for further analysis, leaving behind many nodes that were a replicate of the same compound and compounds that were only produced in traces within the extracts, therefore providing a more realistic view of the *Micromonospora* production. The raw data were preprocessed by MZmine 2 [21] and then uploaded to the GNPS platform to perform the analysis. A metadata file was also uploaded to describe the properties of each sample-strain (taxonomy, geography, ecology). The results were visualized using the Cytoscape software (version 3.8.2) and are publicly available at: https://gnps.ucsd.edu/ProteoSAFe/status.jsp?task=2dfa24f8cd0c404eb980cef3580d3eac, accessed on 2 December 2022.

The FBMN analysis resulted in 447 nodes (parent ions) that were organized in 15 clusters (molecular families, at least two nodes connected). The four main clusters were putatively identified. Two of them were created by the siderophores, one by the phocoenamicins and one by maklamicin and related compounds (Figure 8a). Many of the nodes remained unannotated, highlighting the need for further study of the potential of these strains. Some of them that were related to known compounds were tentatively characterized as new according to the manual dereplication process (see Section 2.6).

Focusing on the phocoenamicins cluster, the three phocoenamicins were putatively detected clustering together (A–C, Figure 8b). As shown in Figure 8b, the cluster was composed of 18 nodes in total. Some of the nodes represented adducts of the same molecule and this was taken into account upon their tentative identification, based on the RT span and the MS/MS fragments, resulting in 10 different compounds. The nodes are marked by experimental *m*/*z* values and colored according to the geographic distribution of the strains that produced them. As it is shown, they are widely distributed in almost all continents. Moreover, taking into account that the FBMN analysis provides relative quantification, in the culture medium RAM2-P V2, phocoenamicins B and C were produced more by the marine-derived strains isolated from the Canary Islands, while phocoenamicin is produced more by the rest of the strains, in agreement with the previously mentioned observations. Furthermore, some nodes could not be identified, indicating that they could putatively represent new analogues of this family of compounds. For example, the parent ion with *m*/*z* 1120.4868 that was directly connected with phocoenamicin C showed a difference of 16 Da, indicating a possible oxygenation and the proposed molecular formula C_56_H_75_ClO_20_ generated for the compound was in agreement (see Section 2.6).

### 2.6. Putative Identification of Possible New Compounds

The LC-UV-HRMS analysis and the manual dereplication tentatively detected several peaks corresponding to compounds with molecular formulae that were not found within various databases, therefore are putatively new metabolites related to the phocoenamicins, the maklamicin or the siderophores.

In particular, the annotations were based on GNPS, searching the compounds that were in each molecular cluster and combining the manual putative dereplication of *m*/*z*, UV profile, RT and proposed molecular formula against the databases previously mentioned (DNP, COCONUT, NPASS and ChemSpider). Table 3 summarizes the putatively unknown compounds grouped by family. Their *m*/*z* values and adduct ions detected, theoretical masses and deviation in ppm, the predicted molecular formulae and the chromatographic retention time are shown. As in Table 2, due to the detection of these molecules in multiple analytical runs, representative values of *m*/*z* and retention times for each compound are displayed.

Several phocoenamicin derivatives were tentatively detected. Some of the molecular formulae predicted (C_48_H_70_O_16_, C_48_H_70_O_17_, C_36_H_50_O_9_, C_42_H_60_O_12_) proposed a possible change in the oligosaccharide motif of phocoenamicins that was not observed before within the family. Spirotetronates have exhibited a huge structural variety and it has been demonstrated that even a small change in the structure of a spirotetronate can result in changing its bioactivity [45]. Therefore, large-scale fermentation should be performed in order to isolate these compounds, elucidate their structure and evaluate their potential bioactivity. 

Maklamicin [29] is the only member of the family naturally produced so far. 29-deoxymaklamicin, that tentatively matched the accurate mass (508.3169) and the proposed molecular formula (C_32_H_44_O_5_) of one of these putatively new compounds, was also reported before, but as a product of a genetically engineered strain [46]. The GNPS analysis revealed five assumably new analogues of maklamicin in total and further research should be carried out to explore the full potential of this family. Likewise, various siderophore-related and putatively new compounds were detected in agreement with the literature reporting that the fountain of actinomycetes producing new siderophores is infinite [47].

## 3. Discussion

The microbial strains explored in this study as possible phocoenamicins producers belonged to the *Micromonospora* genus, with *M. terminaliae* and *M. endophytica* getting assigned more than half of all the strains and it can be proposed that the biosynthesis of the phocoenamicins is restrained within *Micromonospora* sp. Despite the close taxonomic relation of the strains, they displayed great ecological and geographical diversity. Even though the phocoenamicins were originally isolated from marine actinomycetes, the producers identified in this study were isolated from very diverse ecosystems, resulting in 24 terrestrial strains and only three marine-derived strains. Regarding their geographical origin, they were widely distributed throughout the planet, covering almost all continents, confirming the hypothesis that this family of compounds is highly conserved within the *Micromonospora* community.

Using an OSMAC approach, the 27 strains were cultivated in 10 different media each, resulting in 270 fermentations, whose crude extracts were analyzed by LC-HRMS and explored by metabolomics analysis and molecular networking. All 27 strains produced the three known phocoenamicins in at least one of the fermentation media. The manual exploration of the data suggested that phocoenamicin was the most abundant compound, followed by phocoenamicin B and C. Furthermore, the analysis of the production of the three phocoenamicins suggested that, in general for all the 27 strains, the most suitable culture media were RAM2-P V2, FR23, M016 and SAM-6. The influence of the different strains, taxonomic species, culture media, ecological source and geography was studied and the only parameter that could provoke a significant change in the metabolic profile was the culture medium.

The potent RAM2-P V2 medium was selected for further analysis by LC-MS/MS. The three marine-derived strains had a similar chemical response to the medium in comparison to the terrestrial ones. As a result, the relative peak intensities of the three phocoenamicins were analyzed comparing the marine and terrestrial strains. Phocoenamicin was produced more within the terrestrial strains (maximum levels from the strains CA-184181 and CA-238377), while phocoenamicins B and C within the marine strains (CA-214658 and CA-218877). 

The crude extracts of the 270 fermentations were tested against methicillin-resistant *S. aureus* (MRSA) MB5393, *M. bovis* BCG and *M. tuberculosis* H37Ra to evaluate the antimicrobial potential of the strains. All 27 strains exhibited some type of antimicrobial activity under some of the conditions and it was observed that the medium in which the *Micromonospora* strains were grown was clearly a more determining factor than the different strains themselves. The results revealed a rich antimicrobial potential within the 27 strains that should be further investigated.

The combination of the LC-UV-HRMS analysis, metabolomic profiling and molecular networking allowed the putative dereplication of the main compounds produced within the 27 strains in the different growing conditions. The comparative analyses led to the tentative identification of multiple, chemically diverse compounds that were divided into three main groups, the spirotetronates, the siderophores and various other compounds.

The spirotetronate maklamicin, structurally related to phocoenamicins, was also tentatively identified to be produced by all 27 strains. Within the 270 extracts of this study, the production of maklamicin conditioned the production of the phocoenamicins in all cases. The coexistence of maklamicin and the phocoenamicins in the extracts analyzed confirmed the obvious common biosynthetic origin of both families of compounds.

Apart from the spirotetronates, many of the compounds putatively dereplicated belong to the hydroxamate siderophores, as well as various other compounds that were tentatively identified within the extracts with the most abundant being the isoflavones daidzein, genistein and glycitein. This wide and diverse metabolic profile of these strains highlights the need for further research.

Lastly, the analyses tentatively revealed the presence of various peaks of compounds with predicted molecular formulae that were not found in the literature, putatively new metabolites related to the phocoenamicins (8 analogues), the maklamicin (5 analogues) or siderophores (6 analogues) and thus candidates for isolation and discovery of new bioactive compounds. Large-scale fermentation should be performed in order to isolate these compounds, elucidate their structure and evaluate their potential bioactivity.

In this study, we have clearly demonstrated that the combination of manual exploration, metabolomics analysis and molecular networking can be a rapid and efficient way to prioritize strains and further research should be carried out to unveil the chemical diversity of these promising *Micromonospora* strains. Additionally, our research reveals a wide geographical distribution of phocoenamicin producing microbial strains, including both marine and terrestrial strains, opposed to initial statements about phocoenamicin production only in unique marine environments. Our findings have also identified marine-derived *Micromonospora* species as the best producers of phocoenamicins in terms of both the abundance in their extracts of some major members of the structural class and in the variety of molecular structures produced.

## 4. Materials and Methods

### 4.1. Taxonomical Identification of the Producing Microorganisms

The high-performance liquid chromatography coupled to mass spectrometry (HPLC-UV-MS) profile of the phocoenamicins, including RT, UV spectrum and positive and negative mass spectra, was used to search an internal repository of LC/UV/MS analytical data corresponding to 74.647 extracts originated from 16.366 prokaryotic strains (95.6% actinomycetes). This search identified 27 microbial strains belonging to the MEDINA culture collection that potentially biosynthesized at least one of the phocoenamicins.

Genomic DNA from the 27 strains was isolated as previously described [48]. The 16S rRNA gene was PCR-amplified employing the universal eubacterial primers fD1 (50-AGAGTTTGATCCTGGCTCAG-30) and rP2 (50-ACGGCTACCTTGTTACGACTT-30). PCR reactions were carried out in a final volume of 50 μL containing 2 μL of dNTPs (10 mM each), 2 μL of each of the primers (10 μM), 2 μL of the DNA dilution, 5 μL of PCR buffer (10×), and 0.4 μL of Taq Polymerase (5 U/μL). Amplifications were performed on an iCycler iQ^TM^ Real-Time PCR Detection System (Bio-Rad Laboratories, Inc, Hercules, CA, USA).

The PCR products were purified and sequenced at Secugen S. L. (Madrid, Spain). For each product the two strands were sequenced employing the primers mentioned above. The resulting DNA sequence lectures were aligned and visually inspected with Bionumerics 6.6 (Applied Maths, Sint-Martens-Latem, Belgium).

### 4.2. Fermentation of the Producing Microorganisms and OSMAC Approach

Τhe fermentation for each of the 27 strains was generated as follows: a seed culture of the strain was obtained by inoculating a 25 × 150 mm tube containing 10 mL of ATCC-2 medium (potato starch 2 g/L, dextrose 1 g/L, NZ Amine Type E 5 g/L, meat extract 3 g/L, peptone 5 g/L, yeast extract 5 g/L, calcium carbonate 1 g/L, pH 7) with 0.5 mL of a freshly thawed inoculum stock of the producing strain. The tubes were incubated in a rotary shaker at 28 °C, 70% relative humidity and 220 rpm for about 7 days (Kühner incubator model ISF-4-V, Adolf Kühner AG, Brisfelden, Switzerland). 

The fresh inoculum thus generated was mixed and employed to inoculate (5% *v*/*v*) EPA vials (28 × 90 mm glass tubes) each containing 10 mL of fermentation medium. The fermentation media used were as follows: APM9, DEF-15, DEF-15S, DNPM, FPY-12, FPY-2, FR23, M016, RAM2-P V2 and SAM-6. Their composition is summarized in Appendix A. In parallel, a negative control of 10 uninoculated culture media was performed to ensure that any compounds of interest are produced by the microorganisms. All EPA vials were incubated in a rotary shaker at 28 °C, 70% relative humidity, and 220 rpm for 14 days before harvesting (Kühner ISF-4-V incubator, Adolf Kühner AG, Brisfelden, Switzerland).

### 4.3. Extraction of the Fermentation Broths and Analysis by LC-HRMS

The fermentation broths were extracted by adding 10 mL of acetone to each 10 mL fermentation, including the negative control media. The mixtures were then vortexed and shaken in an orbital shaker incubator (Künher ISF-1-W incubator, Adolf Kühner AG, Brisfelden, Switzerland) at 200 rpm for 1 h, centrifuged for 10 min for the separation of the mycelium, and 12 mL of the supernatant were transferred to 16 mm TurboVap^®^ tubes (Biotage, Uppsala, Sweden) to which 600 μL of dimethyl sulfoxide (DMSO) were added. Finally, evaporation of the samples was carried out in a hot nitrogen flow cabinet, until a final volume of 3.0 mL to reach a final concentration of 2 × WBE (Whole Broth Equivalent, that is, the natural concentration of each metabolite in the culture broth) in 20% DMSO in water. The contents of the tubes were transferred to 0.8 mL 96-Well Deep Well plates (Abgene^TM^, Waltham, MA, USA) using the MultiPROBE^®^ II HT robotic liquid handling system (Packard Bioscience Co, Meriden, CO, USA), transferring 540 μL to each well in the plate.

The extracts obtained (as well as the control media) were subjected to liquid chromatography-high resolution mass spectrometry (LC-HRMS) and furthermore tandem mass spectrometry for the extracts obtained from the RAM2-P V2 medium. ESI-TOF and MS/MS spectra were acquired using a Bruker maXis QTOF (Bruker Daltonik GmbH, Bremen, Germany) mass spectrometer coupled to an Agilent 1200 LC (Agilent Technologies, Waldbronn, Germany) with a standard 10-min reversed-phase gradient chromatographic run, as described before [49].

### 4.4. Multivariate Data Analysis (MVDA) with MZmine 2 and MetaboAnalyst 5.0

The LC-MS raw data were initially converted into mzXML files using MSConvert (ProteoWizard, http://proteowizard.sourceforge.net, accessed on 2 December 2022). The converted datasets were imported into MZmine v2.53, a bioinformatics tool for differential analysis of mass spectrometry data [21]. Peak detection was achieved by noise removal, chromatogram construction, and peak deconvolution. Firstly, the mass values were detected using the centroid mode in each spectrum and the noise level of the peaks was set to 100. Then, chromatograms were constructed for each of the mass values using the ADAP chromatogram builder [50], where group intensity threshold and minimum highest intensity were set to 300 and *m*/*z* tolerance to 20 ppm. Next, the “local minimum search” deconvolution algorithm was applied to each constructed chromatogram of each mass ion to detect the individual peaks. For this algorithm, the chromatographic threshold was set to 10%, the search minimum in RT range to 0.2 min, the minimum relative height to 10%, the minimum absolute height to 100 and the minimum ratio of peak top/edge to 2. The separated peaks were then deisotoped using the function of isotopic peaks grouper in which *m*/*z* tolerance was set to 20 ppm, RT tolerance to 0.5 absolute (min), maximum charge of 3 and representative isotope as the most intense. The remaining peaks in different samples were aligned based on the mass and RT of each peak, creating a peak list. The ion *m*/*z* tolerance for alignment was set to 20 ppm, while RT to 0.5 min, and weight for *m*/*z* and RT were 80 and 20, respectively. Finally, the resulting peak list was gap-filled with missing peaks using intensity tolerance of 10%, *m*/*z* tolerance of 20 ppm and RT tolerance of 0.5 min.

At the same time, using the MZmine’s setting of sample parameters, the samples were organized in groups, according to their taxonomy, geography, ecology, culture medium and the different strains. In the end of the above processing, the resulting data were converted into a MetaboAnalyst-CSV (comma-separated values) file, a text file that allows data to be saved in a table-structured format, taking into account the grouping parameters. As a result, five .csv files were generated, one for each of the grouping parameters to study their influence in the metabolic profile obtained. Each of these files was uploaded to the MetaboAnalyst 5.0 platform [22] to carry out the statistical analysis.

The MetaboAnalyst module of the One Factor-Statistical Analysis was used, and each file was uploaded as peak intensities data type and samples in columns (unpaired) format. Missing values were replaced by 1/5 of min positive values of their corresponding values and further feature filtering based on interquantile range (IQR) reduced the dataset to 2500 features. Normalization was performed using total area sums, Pareto scaling, and log transformation to transform the data matrix into a more Gaussian-type distribution and make individual features more comparable. Using the above parameters, PCA and PLS-DA analyses were generated to explore the metabolomics production. Within the PLS-DA analysis, the variables importance in projection (VIP) were studied, and cross validation and permutation test (separation distance (B/W), 100 permutation numbers) were performed.

Furthermore, the LC-MS/MS raw data of the 27 strains cultured in the RAM2-P V2 medium were processed by MZmine 2, employing the same parameters and noise level of 50 for the MS2 level. The processed data were used to visualize the production of the phocoenamicins by the peak intensities plot. Finally, after uploading the MetaboAnalyst file created and using the ecology of the strains as the grouping parameter, the HCA dendrogram was generated.

### 4.5. High-Throughput Screening Assay of the Fermentation Extracts for Antimicrobial Activities

The crude extracts of the 270 fermentations were tested against the growth of methicillin-resistant *S. aureus* (MRSA) MB5393, *M. tuberculosis* H37Ra and *M. bovis* BCG. For the preparation of the inocula and the HTS assays, previously described methodologies were followed [51,52]. 

Briefly, 90 μL/well of the diluted inocula were mixed with 10 μL/well of extracts (final concentration of 0.2 × WBE of each extract in the assay). A dose–response curve of a reference standard compound (vancomycin for MRSA assay and streptomycin for both *Mycobacterium* assays) was included as internal assay plate control. The assays were performed in duplicate for each microorganism.

For MRSA assay, absorbance at OD 612 nm was measured at T_0_ (zero time) and immediately after that, plates were statically incubated at 37 °C for 20 h. After this period, the assay plates were shaken using the DPC Micromix-5 and once more the absorbance at OD 612 nm was measured at Tf (final time). For *Mycobacterium* assays, plates were incubated for 7 days at 5% CO_2_ and 95% humidity and 37 °C. After this incubation, 30 μL of 0.02% resazurin and 15 μL of Tween 20 were added to each well, incubated 24 h and assessed for color development. A change from blue to pink indicates reduction of resazurin and therefore bacterial growth. The wells were read for color change and the data were quantified by measuring fluorescence (excitation 570 nm, emission 615 nm). Both readouts absorbance and fluorescence were measured using an EnVision^®^ multimode plate reader (Perkin Elmer, Waltham, MA, USA). 

In order to process and analyze the data and calculate the RZ’ factor (which predicts the robustness of an assay), the Genedata Screener software (Genedata, Inc., Basel, Switzerland) was employed. For an assay to be accepted, the RZ’ factor must be close to 1 [53]. 

An extract was considered active when the percentage of inhibition was greater than 50%. In all experiments performed in this work, the RZ’ factor obtained was between 0.85 and 0.95. 

### 4.6. Molecular Networking

#### 4.6.1. Classical Molecular Networking

The LC-MS/MS raw data were converted into mzXML files using MSConvert (ProteoWizard, proteowizard.sourceforge.net), uploaded to the Global Natural Products Social Molecular Networking (http://gnps.ucsd.edu, accessed on 2 December 2022) platform using WinSCP (https://winscp.net/eng/index.php, accessed on 2 December 2022) and then processed by GNPS to generate a MS/MS molecular network. The molecular network was created using the online workflow (https://ccms-ucsd.github.io/GNPSDocumentation/, accessed on 2 December 2022) on the GNPS website (http://gnps.ucsd.edu, accessed on 2 December 2022) [54]. The parameters are publicly available at: https://gnps.ucsd.edu/ProteoSAFe/result.jsp?task=37e734c22f5f43cab7f8983d5121cea6&view=written_description, accessed on 2 December 2022.

#### 4.6.2. Preprocessing by MZmine 2 and Feature-Based Molecular Networking

The data files previously converted to mzXML format were imported to MZmine v2.53 for preprocessing [21]. For each sample, the mass detection was set to a noise level of 100 for the MS1 level and 50 for MS2 levels. The chromatograms were built using the ADAP chromatogram builder [50] with a minimum group size set to 5 scans, group intensity threshold and minimum highest intensity set to 300 and *m*/*z* tolerance to 20 ppm. The chromatograms were deconvoluted with the local minimum search algorithm (chromatographic threshold of 10%, search minimum in RT range of 0.1 min, minimum relative height of 10% and minimum absolute height of 100). Deisotoping of the chromatograms was achieved by the isotope peak grouper algorithm with *m*/*z* tolerance set to 20 ppm and RT tolerance to 0.5 min. All samples were combined in a peak list using the join aligner algorithm. The data were filtered in order to keep only peaks with minimum 2 peaks in a row and minimum 2 peaks in an isotope pattern. Lastly, the peak list was gap-filled with an intensity tolerance of 10%, *m*/*z* tolerance of 20 ppm and RT tolerance of 0.5 min. The final peak list was exported in two different format files (.csv and .mgf).

The resulting files were uploaded to GNPS for feature-based molecular networking (FBMN) analysis. A metadata file was also created and uploaded as a text file, describing the properties of each sample-strain (taxonomy, geography, ecology). A molecular network was created with the feature-based molecular networking (FBMN) workflow [44] on GNPS (https://gnps.ucsd.edu, accessed on 2 December 2022, [53] with the following parameters: The data were filtered by removing all MS/MS fragment ions within +/−17 Da of the precursor *m*/*z* and the MS/MS spectra were window filtered by choosing only the top 6 fragment ions in the +/−50 Da window throughout the spectrum. The precursor ion mass tolerance was set to 1 Da and the MS/MS fragment ion tolerance to 0.5 Da. In the molecular network, the edges had a cosine score above 0.6 and more than 6 matched peaks. Edges between two nodes were kept only if each of the nodes appeared in each other’s respective top 10 most similar nodes and the maximum size of a molecular family was set to 100. Then, the spectra in the network were searched against GNPS spectral libraries [54,55] and the library spectra were filtered in the same manner as the data results. The molecular network generated was analyzed and visualized using Cytoscape software (version 3.8.2) [56]. The number of nodes and edges, as well as the average number of neighbors were generated using the Network Analyzer in Cytoscape software.

### 4.7. Dereplication Process and Identification of Putative New Analogues

The LC-UV-HRMS analyses were carried out as previously reported [49]. The major peaks in each chromatogram were initially searched manually against the in-house MEDINA’s LC-HRMS library. Then, the proposed molecular formula, the accurate mass and the UV profile were compared against the Dictionary of Natural Products (DNP), the COlleCtion of Open Natural ProdUcTs (COCONUT), the Natural Product Activity & Species Source database (NPASS) and the ChemSpider database. Additional information was taken into account, such as the biological source and the structure of the compound, where available within these databases. The combination of this manual comparative search along with the metabolomic profiling and molecular networking allowed the putative dereplication of the main compounds produced within the 27 strains in the different growing conditions.

Furthermore, this manual dereplication tentatively detected various peaks of compounds with molecular formulae that were not found within the above-mentioned databases, therefore are putatively new metabolites related to the phocoenamicins, maklamicin or siderophores. These annotations were also based on GNPS, searching the compounds that were in each molecular cluster and combining all the data obtained.

## Figures and Tables

**Figure 1 marinedrugs-20-00769-f001:**
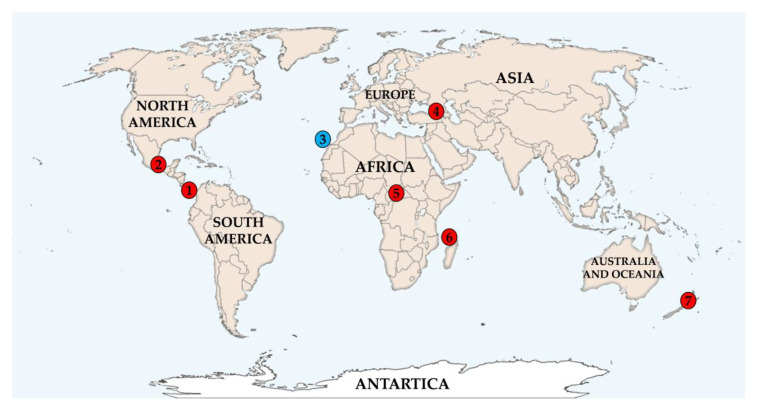
Geographic distribution of the 27 strains within the seven different countries worldwide. The blue colored circle corresponds to the three marine-derived strains and the red to the rest of the terrestrial strains. Numbers correspond to the different countries, regions, approximate Global Positioning System (GPS) coordinates and year of collection: 1. Costa Rica (Guanacaste, 11°2′12.80″ N 85°27′10.84″ W, 1998) 2. Mexico (Las Tuxtlas, Veracruz, 18°28′28.78″ N, 95°12′54.43″ W, 2001) 3. Spain (Gran Canaria, 27°44′42.04″ N, 15°37′42.28″ W, 2004) 4. Georgia, a. (Batumi, Adjara, 41°36′32.94″ N, 41°39′1.51″ E, 2004) b. (Poti, Samegrelo, 42°10′16.14″ N, 41°40′47.06″ E, 2005) c. (Khulo, Adjara, 41°38′51.94″ N, 42°18′51.74″ E, 2005) 5. Central African Republic (Damara/Sibut, 5°19′22.48″ N, 18°57′38.99″ E, 2005) 6. Union of the Comoros, a. (Mdji Diakagnoa, 11°53′2.65″ S, 43°24′44.32″ E, 2005) b. (Ouralé-Ouandaoé, 11°51′5.98″ S, 43°25′46.85″ E, 2005) c. (Tsinimouachongo, 11°50′38.11″ S, 43°22′35.29″ E, 2005) d. (Zikaledjou, 11°48′25.45″ S, 43°16′47.75″ E, 2005) 7. New Zealand (Pehitawa Forest Reserve, 38°21′26.21″ S, 175°11′35.88″ E, 2006).

**Figure 2 marinedrugs-20-00769-f002:**
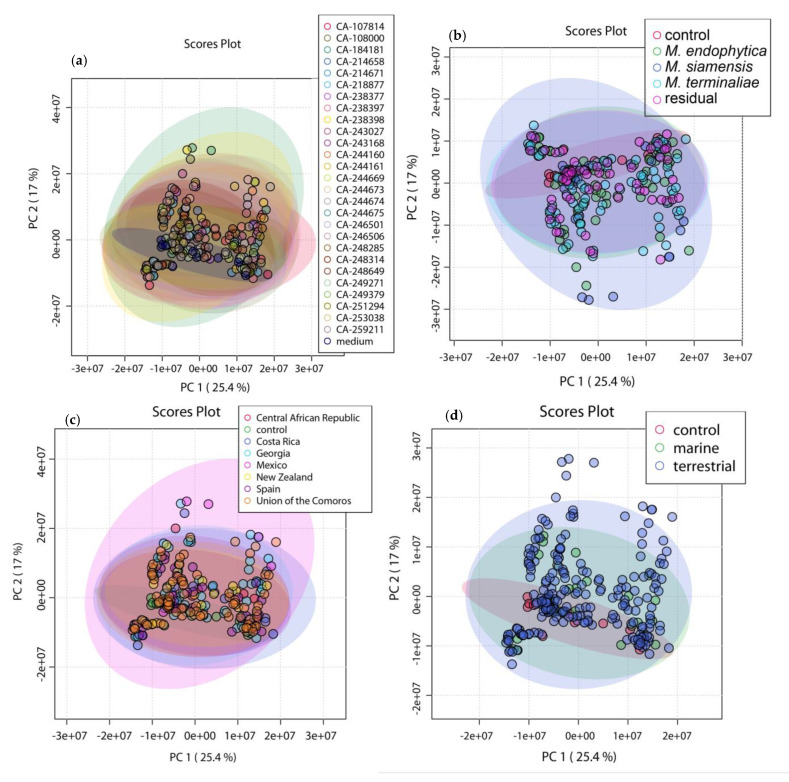
PCA 2D-Score plot of principal component 1 (PC1) (variance of 25.4%) and PC2 (variance of 17%), with a total variance of 42.4%. Five different models were tested, using the parameters: (**a**) different strains; (**b**) taxonomy species; (**c**) geographic origin; (**d**) ecology; and (**e**) culture medium. Only model (**e**) managed to separate the samples and distinguish them in groups according to the culture medium.

**Figure 3 marinedrugs-20-00769-f003:**
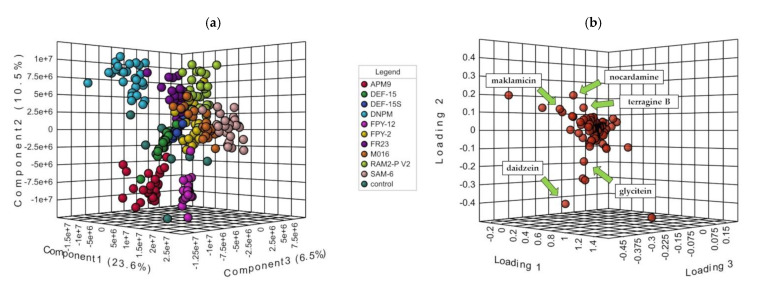
(**a**) PLS-DA 3D-Score plot of the 270 extracts grouped by culture medium and the 10 control extracts where the extracts are clustered according to the culture media; and (**b**) the main compounds that were responsible for the separation. The components that were further from the center of the plot contributed the most to the separation. Only the ones that accounted for Variable Importance in Projection (VIP) higher than 1.5 were considered.

**Figure 4 marinedrugs-20-00769-f004:**
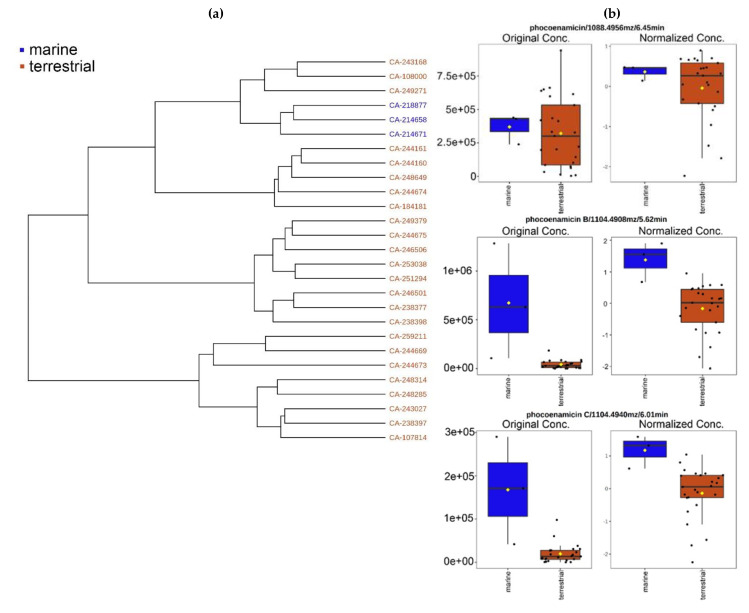
(**a**) Dendrogram generated from the Hierarchical Cluster Analysis showing the close clustering of the three marine-derived strains (blue) compared to the terrestrial (brown); and (**b**) the relative concentration of the three phocoenamicins, before and after normalization techniques and based on their peak intensities and grouped by their source, marine (blue) and terrestrial strains (brown).

**Figure 5 marinedrugs-20-00769-f005:**
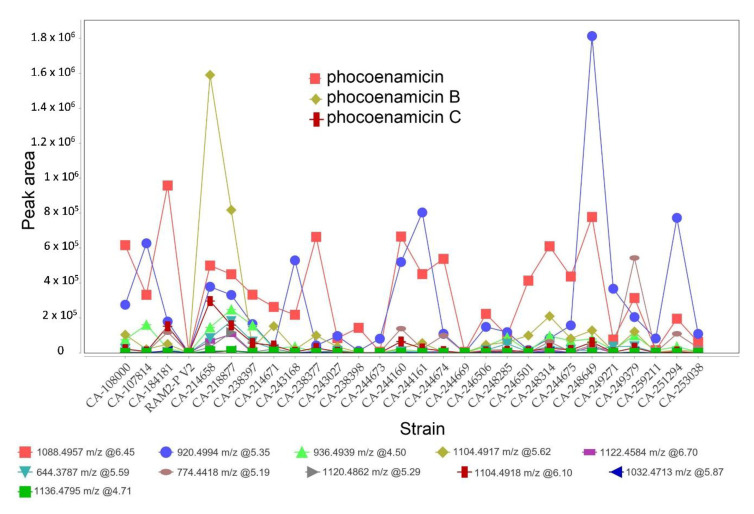
Peak intensity plot based on peak area generated by MZmine 2 for the three phocoenamicins along with the putative phocoenamicin analogues identified in RAM2-P V2 medium. The *m*/*z* values, RT and their corresponding colors-shapes are shown for each compound.

**Figure 6 marinedrugs-20-00769-f006:**
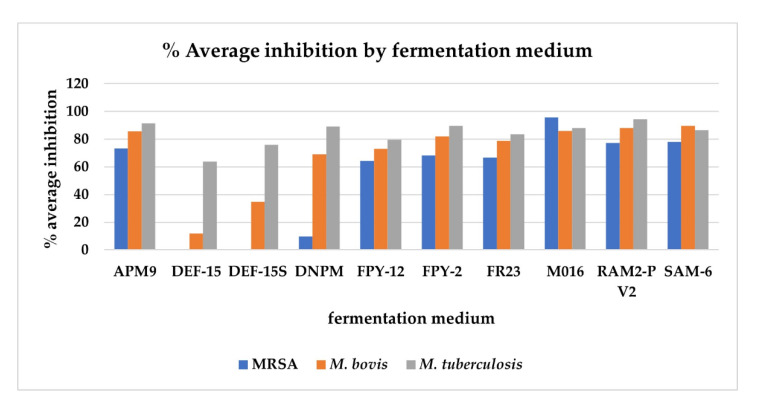
The average of the percentages of inhibition (%) obtained against methicillin-resistant *Staphylococcus aureus* (MRSA) MB5393, *Mycobacterium bovis* BCG and *Mycobacterium tuberculosis* H37Ra from each of the extracts from the 27 strains versus the different media in which they were grown. Extracts were tested at a concentration of 0.2 × WBE.

**Figure 7 marinedrugs-20-00769-f007:**
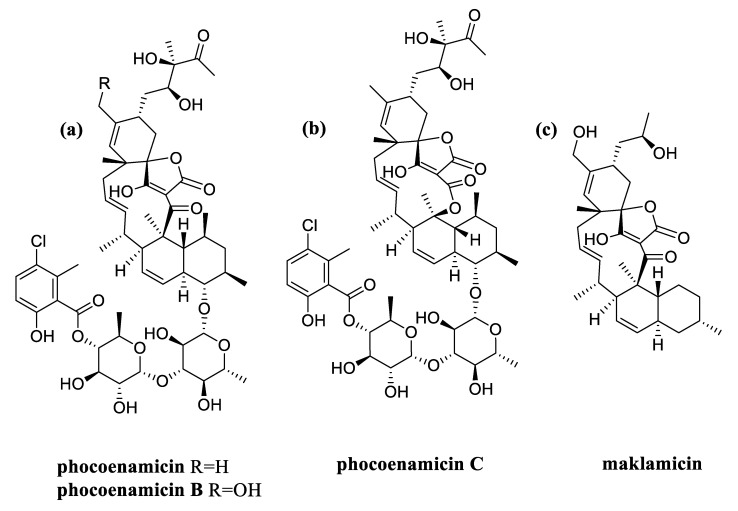
Structures of: (**a**) phocoenamicin and phocoenamicin B; (**b**) phocoenamicin C; and (**c**) maklamicin.

**Figure 8 marinedrugs-20-00769-f008:**
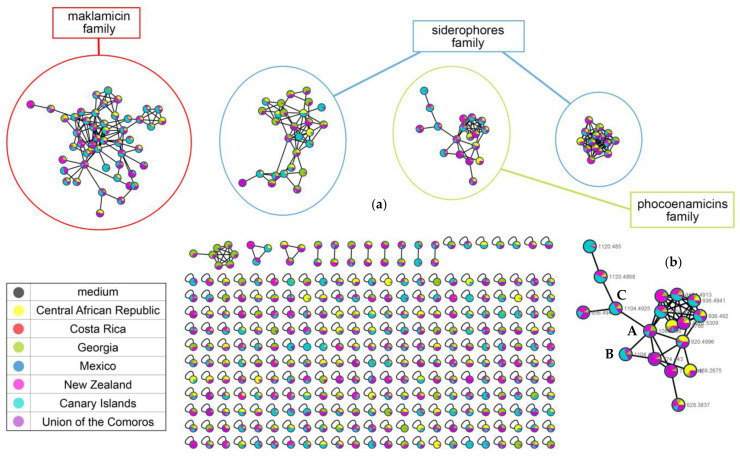
(**a**) The molecular network generated from the FBMN analysis of the 27 strains grown in the RAM2-P V2 medium. The medium was used as control to remove the compounds that were not produced by the strains. The nodes are colored according to the geographic distribution of the strains that produced them. Four main clusters were detected, two created by the siderophores, one by maklamicin and one more from the phocoenamicins; and (**b**) the phocoenamicins cluster revealed 18 nodes and the phocoenamicins were putatively identified as phocoenamicin (A), phocoenamicin B (B) and phocoenamicin C (C). *m*/*z* values of new derivatives in the network are detailed in Table 3.

**Table 1 marinedrugs-20-00769-t001:** Closest species assignment using the EzBioCloud database, the (%) percentage of similarity, the source of isolation and the geographic origin of each strain.

Strain	Geographic Origin *	Ecology	*Micromonospora* Species	Similarity (%)
CA-107814	Costa Rica (1)	soil-rice cultivation	*M. terminaliae*	99.48
CA-108000	Costa Rica (1)	soil-rice cultivation	*M. endophytica*	100
CA-184181	Mexico (2)	soil	*M. siamensis*	99.32
CA-214658	Spain (3)	marine cave sediment	*M. endophytica*	99.63
CA-214671	Spain (3)	marine cave sediment	*M. chaiyaphumensis*	99.84
CA-218877	Spain (3)	marine invertebrate *Porifera* sp.	*M. endophytica*	100
CA-238377	Georgia (4.a)	rhizosphere soil of *Pteridium tauricum*	*M. terminaliae*	99.54
CA-238397	Georgia (4.a)	rhizosphere soil of *Pteridium tauricum*	*M. chalcea*	100
CA-238398	Georgia (4.a)	rhizosphere soil of *Pteridium tauricum*	*M. siamensis*	99.32
CA-243027	Georgia (4.a)	rhizosphere soil of *Pteridium tauricum*	*M. terminaliae*	100
CA-243168	Central African Republic (5)	forest organic humid soil	*M. endophytica*	100
CA-244160	Union of the Comoros (6.a)	forest organic dry soil	*M. terminaliae*	99.53
CA-244161	Union of the Comoros (6.a)	forest organic dry soil	*M. terminaliae*	99.47
CA-244669	Union of the Comoros (6.b)	forest organic humid soil	*M. terminaliae*	99.41
CA-244673	Union of the Comoros (6.b)	forest organic humid soil	*M. chalcea*	100
CA-244674	Union of the Comoros (6.b)	forest organic humid soil	*M. endophytica*	99.31
CA-244675	Union of the Comoros (6.b)	forest organic humid soil	*M. terminaliae*	99.49
CA-246501	Georgia (4.a)	rhizosphere soil of *Pteridium tauricum*	*M. terminaliae*	99.53
CA-246506	Georgia (4.a)	rhizosphere soil of *Pteridium tauricum*	*M. aurantiaca*	99.64
CA-248285	New Zealand (7)	swampy soil	*M. soli*	99.07
CA-248314	New Zealand (7)	swampy soil	*M. endophytica*	99.66
CA-248649	Union of the Comoros (6.c)	forest organic soil	*M. terminaliae*	100
CA-249271	Union of the Comoros (6.d)	forest organic soil	*M. endophytica*	99.38
CA-249379	Union of the Comoros (6.a)	forest organic dry soil	*M. terminaliae*	99.54
CA-251294	Union of the Comoros (6.c)	forest organic soil	*M. terminaliae*	99.51
CA-253038	Georgia (4.b)	rhizosphere soil of *Populus canescens*	*M. siamensis*	99.29
CA-259211	Georgia (4.c)	rhizosphere soil of *Ranunculus buhsei*	No data	-

* Places and dates of collection indicated in brackets are specified in Figure 1.

**Table 2 marinedrugs-20-00769-t002:** Putatively dereplicated metabolites detected in the crude extracts of the fermentation broths.

Compound	*m*/*z* Detected, Adduction	Theoretical Mass (Δ ppm)	Molecular Formula	Retention Time
**spirotetronates**				
phocoenamicin	1088.4982, [M + NH_4_]^+^	1088.4980 (+0.2)	C_56_H_75_ClO_18_	6.44
phocoenamicin B	1104.4927, [M + NH_4_]^+^	1104.4929 (−0.2)	C_56_H_75_ClO_19_	5.53
phocoenamicin C	1104.4933, [M + NH_4_]^+^	1104.4929 (+0.4)	C_56_H_75_ClO_19_	6.02
maklamicin	542.3478, [M + NH_4_]^+^	542.3476 (+0.4)	C_32_H_44_O_6_	6.38
**siderophores**				
nocardamine	601.3551, [M + H]^+^	601.3556 (−0.8)	C_27_H_48_N_6_O_9_	1.26
deoxynocardamine	585.3589, [M + H]^+^	585.3606 (−2.9)	C_27_H_48_N_6_O_8_	1.02
demethylenenocardamine	587.3396, [M + H]^+^	587.3399 (−0.5)	C_26_H_46_N_6_O_9_	0.95
terragine D	479.2848, [M + H]^+^	479.2864 (−3.3)	C_24_H_38_N_4_O_6_	2.54
terragine B	319.1652, [M + H]^+^	319.1652 (0.0)	C_17_H_22_N_2_O_4_	2.80
IC 202B	533.3275, [M + H]^+^	533.3293 (−3.4)	C_23_H_44_N_6_O_8_	1.51
deferoxamine	561.3604, [M + H]^+^	561.3606 (−0.4)	C_25_H_48_N_6_O_8_	0.70
proferrioxamine A1	547.3450, [M + H]^+^	547.3450 (0.0)	C_24_H_46_N_6_O_8_	0.65
desferrioxamine D1	603.3695, [M + H]^+^	603.3712 (−1.1)	C_27_H_50_N_6_O_9_	1.06
legonoxamine A	637.3914, [M + H]^+^	637.3919 (−0.8)	C_31_H_52_N_6_O_8_	2.34
legonoxamine G	623.3763, [M + H]^+^	623.3763 (0.0)	C_30_H_50_N_6_O_8_	2.21
legonoxamine H	665.3861, [M + H]^+^	665.3869 (−1.2)	C_32_H_52_N_6_O_9_	2.47
microferrioxamine B	729.5473, [M + H]^+^	729.5484 (−1.5)	C_37_H_72_N_6_O_8_	4.78
microferrioxamine C	743.5632, [M + H]^+^	743.5641 (−1.2)	C_38_H_74_N_6_O_8_	4.97
microferrioxamine D	757.5792, [M + H]^+^	757.5797 (−0.7)	C_39_H_76_N_6_O_8_	5.21
acyl ferrioxamine 2	679.4015, [M + H]^+^	679.4025 (−1.4)	C_33_H_54_N_6_O_9_	2.78
**various**				
daidzein	255.0648, [M + H]^+^	255.0652 (−1.6)	C_15_H_10_O_4_	2.55
genistein	271.0594, [M + H]^+^	271.0601 (−2.6)	C_15_H_10_O_5_	3.20
glycitein	285.0754, [M + H]^+^	285.0757 (−1.1)	C_16_H_12_O_5_	2.67
antascomicin D	646.3946, [M + H]^+^	646.3950 (−0.6)	C_36_H_55_NO_9_	3.80
21-demethyl-leptomycin A	513.3208, [M + H]^+^	513.3211 (−0.6)	C_31_H_44_O_6_	5.07
indothiazinone-4-carboxylic acid	273.0327, [M + H]^+^	273.0328 (−0.4)	C_13_H_8_N_2_O_3_S	2.98
anandin A	358.2743, [M + H]^+^	358.2741 (+0.6)	C_23_H_35_NO_2_	4.52
ganefromycin epsilon	644.3794, [M + H]^+^	644.3793 (+0.2)	C_36_H_53_NO_9_	4.70
actiphenol	276.1238, [M + H]^+^	276.1230 (+2.9)	C_15_H_17_NO_4_	2.48
N^b^-Acetyltryptamine	203.1177, [M + H]^+^	203.1179 (−1.0)	C_12_H_14_N_2_O	2.41
N-Acetyltyramine	180.1023, [M + H]^+^	180.1019 (+2.2)	C_10_H_13_NO_2_	0.92
actiphenamide	294.1331, [M + H]^+^	294.1336 (−1.7)	C_15_H_19_NO_5_	2.48
antibiotic BE 54476	392.2429, [M + H]^+^	392.2431 (−0.5)	C_22_H_33_NO_5_	4.14

**Table 3 marinedrugs-20-00769-t003:** Potential derivatives detected in the crude extracts of the fermentation broths.

Compound	*m*/*z* Detected, Adduction	Theoretical Mass (Δ ppm)	Molecular Formula Predicted	Retention Time
**spirotetronates**				
phocoenamicin derivative	1120.4862, [M + NH_4_]^+^	1120.4878 (−1.4)	C_56_H_75_ClO_20_	5.29
phocoenamicin derivative	1032.4713, [M + NH_4_]^+^	1032.4718 (−0.5)	C_53_H_71_ClO_17_	5.70
phocoenamicin derivative	1122.4584, [M + NH_4_]^+^	1122.4590 (−0.5)	C_56_H_74_Cl_2_O_18_	6.71
phocoenamicin derivative	1136.4795, [M + NH_4_]^+^	1136.4828 (−2.9)	C_56_H_75_ClO_21_	4.54
phocoenamicin derivative	920.4994, [M + NH_4_]^+^	920.5002 (−0.9)	C_48_H_70_O_16_	5.24
phocoenamicin derivative	936.4939, [M + NH_4_]^+^	936.4951 (−1.3)	C_48_H_70_O_17_	4.46
phocoenamicin derivative	644.3787, [M + NH_4_]^+^	644.3793 (−0.9)	C_36_H_50_O_9_	5.55
phocoenamicin derivative	774.4418, [M + NH_4_]^+^	744.4423 (−0.7)	C_42_H_60_O_12_	5.19
maklamicin derivative	509.3250, [M + H]^+^	509.3262 (−2.4)	C_32_H_44_O_5_	7.30
maklamicin derivative	495.3091, [M + H]^+^	495.3105 (−2.8)	C_31_H_42_O_5_	7.09
maklamicin derivative	511.3060, [M + H]^+^	511.3054 (+1.2)	C_31_H_42_O_6_	6.12
maklamicin derivative	539.3350, [M + H]^+^	539.3367 (−3.2)	C_33_H_46_O_6_	6.60
maklamicin derivative	523.3049, [M + H]^+^	523.3054 (−1.0)	C_32_H_42_O_6_	6.91
**siderophores**				
nocardamine derivative	599.3379, [M + H]^+^	599.3399 (−3.3)	C_27_H_46_N_6_O_9_	1.85
deferoxamine derivative	631.4384, [M + H]^+^	631.4389 (−0.8)	C_30_H_58_N_6_O_8_	2.88
deferoxamine derivative	645.4531, [M + H]^+^	645.4545 (−2.2)	C_31_H_60_N_6_O_8_	3.23
deferoxamine derivative	687.4638, [M + H]^+^	687.4651 (−1.9)	C_33_H_62_N_6_O_9_	3.44
deferoxamine derivative	659.4342, [M + H]^+^	659.4338 (+0.6)	C_31_H_58_N_6_O_9_	2.94
deferoxamine derivative	671.4694, [M + H]^+^	671.4702 (−1.0)	C_33_H_62_N_6_O_8_	3.66

## Data Availability

Data availability is indicated through links within the article or included as Appendix A.

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
