# Peer review of "Exploring Micromonospora as Phocoenamicins Producers"

_marinedrugs, 2022, doi:10.3390/md20120769_

Round 1
Reviewer 1 Report
This manuscript by Kokkini et al. analyzed and compared the taxonomy and secondary metabolites of 27 Actinomycete strains which may produce phocoenamicins. The antimicrobial activities of the crude extracts and prediction of novel compounds by LC-UV-HRMS were also investigated. Overall, the manuscript is more like a review paper rather than a research article. Thus, this reviewer recommends the acceptance for publication in Marine Drugs after the following major issues are addressed.
1. In Figure 6, please clarify the test concentration of the crude extracts.
2. In the first paragraph of page 13, the author claimed that “This relation of the two families is not reported before and suggests the existence of a common biosynthetic 355 pathway in the production of the two families that should be further explored”. Looks like they think the production of Maklamicin is unusual. But for most readers, this relation of the two families is supposed to be obvious. This paragraph is nonsense and does not provide anything useful.
3. In Table 3, the differences between accurate mass detected and m/z detected are too big that the reasoning of maklamicin or phocoenamicin derivatives are not convincing. BTW, m/z detected data should be provided also in table 2.
4. In section 2.6 Putative identification of possible new compounds, at least one new compound should be physically isolated and structurally elucidated.
Author Response
This manuscript by Kokkini et al. analyzed and compared the taxonomy and secondary metabolites of 27 Actinomycete strains which may produce phocoenamicins. The antimicrobial activities of the crude extracts and prediction of novel compounds by LC-UV-HRMS were also investigated. Overall, the manuscript is more like a review paper rather than a research article. Thus, this reviewer recommends the acceptance for publication in Marine Drugs after the following major issues are addressed.
We thank this reviewer for the time and effort employed to review our article. Please note that we do not share his/her view about our paper being more a review than a research article. It is true that our research originates from available LC/MS analytical data, obtained through many years of analyses of extracts from our collection, we have performed in our study cultivation of strains using an OSMAC approach, extraction, LC/MS, including low and high resolution MS detection and antimicrobial testing of extracts plus data analysis of a large volume of data generated. In our opinion this cannot be regarded as a review.
- In Figure 6, please clarify the test concentration of the crude extracts.
We thank this reviewer for raising this point which perhaps was not sufficiently clear in the previous version of the article. We define concentration in terms of whole broth equivalent (WBE), that is the natural concentration of secondary metabolites in the culture broth. Our extracts are generated at a concentration of 2×WBE. Please note that we prefer to express concentration in that way rather than drying out extracts and redissolving them at a concentration normalized by weight. WBE allow the direct comparison of the relative production of the target metabolites in the different culture conditions used whereas in the normalization by weight that would not be possible without the application of correction factors associated to the different weight of each extract. We have included the information about WBE concentration in section 4.3 of materials and methods. According to the explanation above and the procedure followed in the antimicrobial testing, the final concentration of extract in those tests will be 0.2 × WBE. This information has been added in section 4.5 of the materials and methods section and also in figure 6 caption.
- In the first paragraph of page 13, the author claimed that “This relation of the two families is not reported before and suggests the existence of a common biosynthetic 355 pathway in the production of the two families that should be further explored”. Looks like they think the production of Maklamicin is unusual. But for most readers, this relation of the two families is supposed to be obvious. This paragraph is nonsense and does not provide anything useful.
We acknowledge this comment from the reviewer. The maklamicin production of course should not be regarded as unusual in the species analyzed, as it is also a spirotetronate produced by Micromonospora strains and the biosynthetic relationship between members of both families is obvious. We have therefore eliminated the sentence on page 13 and we have also modified the sentence on page 17 lines 787-788 of the discussion which now reads: “The coexistence of maklamicin and the phocoenamicins in the extracts analyzed confirms the obvious common biosynthetic origin of both families of compounds”
- In Table 3, the differences between accurate mass detected and m/z detected are too big that the reasoning of maklamicin or phocoenamicin derivatives are not convincing. BTW, m/z detected data should be provided also in table 2.
After this reviewer’s comment, we have realized that the information provided in one of the columns is confusing. The information in the column with the heading “Accurate Mass detected” referred to the experimental Accurate Mass value (M), calculated from the corresponding [M + NH4]+ or [M + H]+ m/z values by subtraction of the masses of NH4+ or H+, respectively. As M is not a common parameter used in mass spectrometry analysis, we have replaced this value in the revised version by the theoretical value of the corresponding adduct, [M + NH4]+ or [M + H]+, for each molecular formula proposed and we have also included the deviation between the experimental and the theoretical value. In all cases such deviation is within the admissible range in HRMS analysis (below 5 ppm).
The same information has also been added in table 2. Please note that tree compounds, namely ornibactin C8, elaiophylin, and nocamycin I, have been eliminated from this table due to differences between the theoretical and the experimental mass higher than 5 ppm that make questionable their identification. References citing these compounds in the text have also been eliminated.
In section 2.6 Putative identification of possible new compounds, at least one new compound should be physically isolated and structurally elucidated.
The isolation of four of these new compounds, one maklamicin and three phocoenamicin derivatives has already been performed from scaled-up cultures of the marine derived strains (CA-214671, CA-214658 and CA-218877). The preparation of a second research article describing their structural elucidation and biological properties is in progress. When considering the possibility of including that information in this article we realized that it would result in an unusually long and dense article having around 45-50 pages. For this reason, we decided to split the information into two different articles. We consider that the information described in this article constitutes an original and complete piece of research that will be complemented with the publication of that second article on the new molecules isolated.
Reviewer 2 Report
Comments on Manuscript marinedrugs-2013771
General Comments
The manuscript is excellently written and gives a complete overview on phocoenamicin producers of the genus Micromonospora. 27 strains of the genus Micromonospora, which were isolated from very diverse ecosystems were all identified as producers of the spirotetronate-type antibiotics maklamicin, phocoenamicin and phocoenamcin B and C. The strains were completely taxonomically characterized, the production of the spirotetronate antibiotics were investigated intensively by the OSMAC method, and their antimicrobial activities against clinically relevant pathogenic bacteria were determined. Besides the known spirotetronates maklamicin and phocoenamicins, potentially new derivatives of maklamicin and phocoenamicin were detected in crude extracts of the fermentation broths, besides known siderophores and new siderophore derivatives. The manuscript can be accepted after minor revision.
Specific Comments
The term ‘actinomycetes‘ should be written within the whole text in small letters.
Part 4.1: Have the authors analyzed all 74.647 extracts originated from 16.366 prokaryotic strains by HPLC/UV/MS for production of phocoenamicins? How many actinomycetes strains were included to the total of 16.366 prokaryotic strains?
References list: The references should be harmonized using small letters citation, as needed for references 1, 3, 7, 11, 16, 17, 24, 26, 29, 30, 31, 38, 39, 44, 47, 51, 52, and 55.
Ref. 13: Change to ‘Actinospica‘.
Ref. 36: Change to ‘Desferrioxamine B‘.
Ref. 29: Change reference to:
Igarashi, Y.; Ogura, H.; Furihata, K.; Oku, N.; Indananda, C.; Thamchaipenet, A. Maklamicin, an antibacterial polyketide from an endophytic Micromonospora sp. J. Nat. Prod. 2011, 74, 670–674.
Author Response
Comments and Suggestions for Authors
Comments on Manuscript marinedrugs-2013771
General Comments
The manuscript is excellently written and gives a complete overview on phocoenamicin producers of the genus Micromonospora. 27 strains of the genus Micromonospora, which were isolated from very diverse ecosystems were all identified as producers of the spirotetronate-type antibiotics maklamicin, phocoenamicin and phocoenamcin B and C. The strains were completely taxonomically characterized, the production of the spirotetronate antibiotics were investigated intensively by the OSMAC method, and their antimicrobial activities against clinically relevant pathogenic bacteria were determined. Besides the known spirotetronates maklamicin and phocoenamicins, potentially new derivatives of maklamicin and phocoenamicin were detected in crude extracts of the fermentation broths, besides known siderophores and new siderophore derivatives. The manuscript can be accepted after minor revision.
We acknowledge the positive comments of this reviewer.
Specific Comments
- The term ‘actinomycetes‘ should be written within the whole text in small letters.
It has been corrected.
- Part 4.1: Have the authors analyzed all 647 extracts originated from 16.366 prokaryotic strains by HPLC/UV/MS for production of phocoenamicins? How many actinomycetes strains were included to the total of 16.366 prokaryotic strains?
Please note that this article takes advantage of the existence at our institution of previous LC/MS analytical data of 74.647 extracts originated from 16.366 prokaryotic strains. These analyses have been performed over several years and the information was stored in a library. This library is available for searches of new molecules of interest, and that is the process that we initially performed with the known phocoenamicins to identify the 27 strains used in the research described in this article. Most of the prokaryotic strains analyzed are actually actinomycetes. We have calculated the percentage and it equals 95.6 %. We have added this information on page 18 line 832, section 4.1 of Materials and Methods.
- References list: The references should be harmonized using small letters citation, as needed for references 1, 3, 7, 11, 16, 17, 24, 26, 29, 30, 31, 38, 39, 44, 47, 51, 52, and 55.
Please note that we usually respect the use of capital/small letters in the titles of the articles according to the format used in the original article. After looking for the above-mentioned articles, their titles were found to be in capital letters in the original publications. We have found that this also seems to be the policy of Marine Drugs, where you can find in published articles the names of articles in references written using both formats. The copy received for revision, edited by the Marine Drugs editorial office also respects both formats. In any case, we are open to adapt the format of our references as required by Marine Drugs.
- 13: Change to ‘Actinospica‘.
It has been corrected.
- 36: Change to ‘Desferrioxamine B‘.
The compound’s name in the article’s title is Deferoxamine B, as it can be found in https://www.mdpi.com/1420-3049/26/11/3255. “Desferrioxamine“ is a synonym for the same compound. The letter B in the title was capitalized.
- 29: Change reference to:
Igarashi, Y.; Ogura, H.; Furihata, K.; Oku, N.; Indananda, C.; Thamchaipenet, A. Maklamicin, an antibacterial polyketide from an endophytic Micromonospora sp. J. Nat. Prod. 2011, 74, 670–674.
Some of the authors‘ names were missing and it has been corrected.
Round 2
Reviewer 1 Report
No more suggestions.
Author Response
We acknowledge the effort taken to revise our article again and the positive outcome of such revision